# Effect of Lactic Acid Bacteria on the Pharmacokinetics and Metabolism of Ginsenosides in Mice

**DOI:** 10.3390/pharmaceutics13091496

**Published:** 2021-09-17

**Authors:** Ji-Hyeon Jeon, Jaehyeok Lee, Jin-Hyang Park, Chul-Haeng Lee, Min-Koo Choi, Im-Sook Song

**Affiliations:** 1BK21 FOUR Community-Based Intelligent Novel Drug Discovery Education Unit, Vessel-Organ Interaction Research Center (VOICE), Research Institute of Pharmaceutical Sciences, College of Pharmacy, Kyungpook National University, Daegu 41566, Korea; kei7016@naver.com (J.-H.J.); here0723@gmail.com (J.L.); wlsgid1957@naver.com (J.-H.P.); 2College of Pharmacy, Dankook University, Cheon-an 31116, Korea; hang1130@naver.com

**Keywords:** red ginseng extract (RGE), lactic acid bacteria (LAB), ginsenoside metabolism, pharmacokinetics

## Abstract

This study aims to investigate the effect of lactic acid bacteria (LAB) on in vitro and in vivo metabolism and the pharmacokinetics of ginsenosides in mice. When the in vitro fermentation test of RGE with LAB was carried out, protopanaxadiol (PPD) and protopanaxadiol (PPD), which are final metabolites of ginsenosides but not contained in RGE, were greatly increased. Compound K (CK), ginsenoside Rh1 (GRh1), and GRg3 also increased by about 30%. Other ginsenosides with a sugar number of more than 2 showed a gradual decrease by fermentation with LAB for 7 days, suggesting the involvement of LAB in the deglycosylation of ginsenosides. Incubation of single ginsenoside with LAB produced GRg3, CK, and PPD with the highest formation rate and GRd, GRh2, and GF with the lower rate among PPD-type ginsenosides. Among PPT-type ginsenosides, GRh1 and PPT had the highest formation rate. The amoxicillin pretreatment (20 mg/kg/day, twice a day for 3 days) resulted in a significant decrease in the fecal recovery of CK, PPD, and PPT through the blockade of deglycosylation of ginsenosides after single oral administrations of RGE (2 g/kg) in mice. The plasma concentrations of CK, PPD, and PPT were not detectable without change in GRb1, GRb2, and GRc in this group. LAB supplementation (1 billion CFU/2 g/kg/day for 1 week) after the amoxicillin treatment in mice restored the ginsenoside metabolism and the plasma concentrations of ginsenosides to the control level. In conclusion, the alterations in the gut microbiota environment could change the ginsenoside metabolism and plasma concentrations of ginsenosides. Therefore, the supplementation of LAB with oral administrations of RGE would help increase plasma concentrations of deglycosylated ginsenosides such as CK, PPD, and PPT.

## 1. Introduction

Red ginseng extract (RGE), one of the most popular herbal medicines, has been investigated for its efficacy. RGE not only reinforces immune function, but also has anti-cancer, anti-diabetes, anti-inflammation, antioxidation, and liver protective effects [1]. Ginseng glycoside, called ginsenoside, is a major active pharmacological component, which causes the efficacy of RGE [2,3,4]. The serum concentrations of GRg1 and GRb1 showed a good correlation with NO releasing effect, a marker for anti-inflammatory effect [5,6]. In addition, the safety, pharmacokinetics, and preliminary efficacy of CK as an anti-rheumatoid arthritis drug are under clinical investigation in China (Study No. NCT03755258) [7]. The plasma exposure of CK showed a linear increase over the oral dose range of 100–400 mg of CK tablet. CK was safe and well-tolerated for CK dose regimen (100–400 mg of CK, once daily for 9 days) [8,9]. The plasma concentration of CK (C_max_ 254.5 ng/mL) in subjects who received 3 g fermented RGE (contains 10.9 mg CK) was substantially higher than that (C_max_ 8.4–24.8 ng/mL) in subjects who received RGE (does not contain CK), suggesting the critical role of fermentation process using lactic acid bacteria (LAB) [10,11]. In a 4-week, randomized, double-blind clinical study of patients with allergic rhinitis, intake of fermented RGE (750 mg/day; contained 45.8 mg CK) alleviated nasal congestion [12]. GRg3 and CK are effective against various human cancer cells as well as in tumor-bearing animal models in a dose-dependent manner [13]. GRg3 suppresses tumor growth and inhibits tumor angiogenesis by inhibiting vascular endothelial growth factor-dependent pathway [14]. GRg3 and CK induce apoptosis and cell cycle arrest at the G0/G1 phase and decreases drug resistance by inactivating nuclear factor kappa light chain enhancer of activated B cells (NF-kB) or a caspase-dependent pathway [15,16]. Administration of GRg3 tablet (20 mg twice daily for 30 days) combined with gemcitabine and cisplatin in patients with advanced esophageal cancer increased the one-year survival rate and reduced drug-associated adverse events [15,17]. Meta-analysis of 20 clinical studies involving GRg3 (20 mg twice daily 6–24 weeks) combined with non-small cell lung cancer therapeutics indicated that Rg3 enhances short-term efficacy, prolongs overall survival rate, and reduces treatment-related toxicity [14,18,19,20]. Plasma exposure of GRg3 was increased 7.3-fold over the 6-fold dose range (10–60 mg, intramuscular injection) without a significant change in clearance and volume of distribution in Chinese healthy volunteers [21].

RGE showed anti-diabetic effect through the suppression of hepatic gluconeogenesis via the activation of adenosine monophosphate-activated protein kinase (AMPK) and peroxisome proliferator-activated receptors-gamma (PPARγ) and through the increased glucose uptake in adipocytes or skeletal muscle cells [22,23]. RGE fermented with *Lactobacillus plantarum*, which is involves in the deglycosylation of PPD-type ginsenosides [24], showed better antidiabetic effects compared with RGE treatment in the streptozotocin-induced diabetic mice [25]. Supplementation of hydrolyzed ginseng extract that are fermented to have higher content of GRg1, GRg3, and CK (960 mg/day for 8 weeks) significantly decreased the levels of fasting plasma glucose and postprandial glucose in 12 impaired fasting glucose participants [26]. These reports suggested that the supplementation of RGE with higher content of pharmacologically active ginsenosides may have better anti-diabetic activity. Similarly, comparative studies on the efficacy of fermented RGE versus RGE were reported. Lee et al. [27] reported the increased anti-wrinkle efficacy, whitening efficacy, and reduced toxicological potency of fermented RGE that contains increased ginsenoside metabolites, such as GRg3, CK, GRh1, and GF2, compared to RGE. Nan et al. [28] reported that fermented ginseng with *Lactobacillus fermentum* ameliorated hyperlipidemia and liver injury induced by a high fat diet through the increased content of GRg3, GRh1, GRh2, and GF2. *Bifidus* fermentation increased the content of GRg3 and GRh2 and it increased hypolipidemic and hypoglycemic effects of RGE in hyperlipidemic mice [29]. These reports suggested that the fermentation of RGE with LAB may change the content of ginsenoside and the efficacy of RGE.

Many studies have focused on the role of LAB in the metabolism of ginsenosides. Ginsenoside has a sugar moiety structure bound to an oleanane and dammarane structure. Depending on the location of the carbon bond of the sugar chain, the type of dammarane–ginsenoside is determined; it is either protopanaxadiol (PPD)-type or PPT-type [20,30,31]. Further subdivisions occur depending on the length of the sugar chain (Figure 1). Belonging to the PPD-type ginsenosides are GRb1, GRb2, GRc, GRd, GRh2, GF2, GRh2, CK, and PPD, whereas GRe, GRg1, GRf, GF1, GRh1, and PPT all belong to PPT-type ginsenosides [20]. Upon human administration, large molecular weight ginsenosides (they have long sugar chains, such as GRb1, GRb2, GRc, GRd, GRe, GRf, and GRg1) undergo deglycosylation by the intestinal microbiota to small molecular weight ginsenosides (have short sugar chains, such as GRh2, CK, PPD, GRh1, GF1, and PPT) [32,33,34]. Therefore, there has been a metabolic relationship in the subdivision of ginsenosides according to the deglycosylation status [32].

The metabolism of ginsenosides is mediated by gut the microbiota, in which the involvement of LAB was reported (Figure 1) [32,35,36,37,38]; *Lactobacillus* sp. [32], *Bifidobacterium* sp. [39,40,41,42,43,44], *Eubacterium* sp., *Fusobacterium* sp., *Bacteroides* sp., and *Microbacterium* sp. [40,42,43,44,45]. More specifically, *L. brevis* is involved in GRd → GF2 → CK metabolism [39,41]. *L. sakei* [24], *L. plantarum* [24], and *Bifidobacterium longum* [46,47] are involved in GRb1, GRb2, GRc → GRd metabolism. *B. longum* was also involved in GRc → GRd metabolism. *Mircobacterium* sp. are involved in GRb1, GRb2 → GRd → GRg3 metabolism [37,38]. *Bifidobacterium* sp., *Eubacterium* sp., and *Bacteroides* sp. are generally involved in the PPD-type and PPT-type metabolism (Figure 1) [40,42,43,44,45].

LAB plays some roles in a part of metabolism or immune system [48]. With an increasing understanding of the beneficial role of LAB, LAB is administered as health supplements and, therefore, the demand for LAB has grown rapidly. This has led to frequent co-administration of LAB and ginseng products. Additionally, based on the beneficial role, Combi-formulation of LAB and ginseng has been introduced in the Korean health supplement market. Moreover, information regarding the interaction between RGE and LAB is limited, although the involvement of LAB in the ginsenoside metabolism has been proven [32,35,36]. As most studies focused on the role of LAB in in vitro ginsenoside metabolism, we investigated the effects of LAB on in vivo ginsenoside metabolism and the resultant pharmacokinetics of ginsenosides. Jeon et al. [31] reported the species difference in the gut metabolism and absorption of ginsenosides in the pathway from GRd to PPD and from GRe to PPT between mice and rats. The pharmacokinetic features of ginsenosides after repeated oral administration of RGE in humans were similar to those in mice rather than those in rats, i.e., GRb1, GRb2, GRc, GRd, GRg3, CK, PPD, and GRe were detected in mouse plasma, but GRb1, GRb2, GRc, GRd, PPD, and PPT were detected in rat plasma after the repeated oral administration of RGE [31]. In human plasma, GRb1, GRb2, GRc, GRd, GRg3, CK, PPD, and PPT were detected [34]. Moreover, the half-lives of GRg3, CK, and PPD in mice were similar to those in human subjects [31,34]. Based on these previous results, we selected mice as experimental animals in this study.

The specific Lactobacillus species, such as *Lactobacillus* sp., and *Bifidobacterium* sp., affect ginsenoside metabolism and are present in the intestine [24,35,36,49]. To further ascertain whether LAB affects the metabolism of ginsenoside, it is necessary to limit the LAB growth, which already exists in the body. Amoxicillin is a penicillin-based antibiotic that works extensively on bacterial infectious diseases and even a small dose of it limits the most of the lactobacillus growth influenced to the metabolism of ginsenoside [50,51,52]. Therefore, the study aimed to investigate the effects of LAB on the in vivo metabolism and pharmacokinetics of ginsenosides after the pretreatment of amoxicillin antibiotics in mice.

## 2. Materials and Methods

### 2.1. Materials

RGE was purchased from the Punggi Ginseng Cooperative Association (Youngjoo, Kyungpook, Korea) (Table 1). 20(S)-ginsenosides Rb1 (GRb1), GRb2, GRc, GRd, GRg1, GRg3, GRe, GRh1, GRh2, GF1, GF2, 20(S)-compound K (CK), 20(S)-proptopanaxadiol (PPD), and 20(S)-protopanaxatriol (PPT) were purchased from the Ambo Institute (Daejeon, Korea). Hank’s balanced salt solution (HBSS, pH 7.4), propranolol, caffeine, atenolol, ofloxacin, metformin, berberine (internal standards, IS), and pooled mouse plasma were purchased from Sigma–Aldrich Chemical Co. (St. Louis, MO, USA). Commercially available lactic acid bacterial formulation (LAB) was purchased from Chong Kun Dang Health Care (Dangjin-si, Chungcheongnam-do, Korea) (Table 2). Difco Lactobacilli MRS broth was purchased from BD BioSciences (Sparks, MD, USA). Collagen-coated 12-transwell, Dulbecco’s modified eagle medium, phosphate-buffered saline (PBS), fecal bovine serum, non-essential amino acids, penicillin-streptomycin, pooled liver microsomes prepared from male CD-1 mice were purchased from Corning Life Sciences (Tewksbury, MA, USA). All other chemicals and solvents were of reagent or analytical grade.

### 2.2. In Vitro RGE Fermentation Study with LAB

For identifying the effects of LAB on ginsenoside metabolism, in vitro incubation study was preceded. Autoclaved RGE (100 mg/10 mL in distilled water) and autoclaved MRS broth (550 mg/10 mL in distilled water) were mixed and added 2 g of LAB (contained 1 billion CFU). Then, the mixture was incubated for 8 days in a shaking incubator at 37 and 300 rpm. For the analysis of ginsenosides, 0.2 mL aliquot of incubation mixture was collected every 24 h for 7 days. The incubation medium was diluted 100-fold with distilled water and stored at −80 °C for the analysis of ginsenosides using a liquid chromatography-tandem mass spectrometry (LC–MS/MS) system. Microbial growth was determined by measuring the optical density at 600 nm (OD_600_) using UV spectrophotometry.

To compare the involvement of LAB on individual ginsenoside, an aliquot (10 μL) of each ginsenoside stock solution (10 mM in methanol) was mixed with autoclaved MRS broth (550 mg/10 mL in distilled water) and added 2 g of LAB (contained 1 billion CFU). Then, the mixture was incubated for 7 days in a shaking incubator at 37 °C and 300 rpm. For the analysis of ginsenosides, 0.2 mL aliquot of incubation mixture was collected every 24 h for 7 days. The incubation medium was diluted 100-fold with distilled water and stored at −80 °C. For the ginsenoside analysis, 200 μL of an IS solution (0.05 ng/mL berberine in methanol) was added to 30 μL of 100-fold diluted samples. The mixture was vortexed for 15 min and centrifuged at 16,000× *g* for 5 min. After centrifugation, a 2 μL aliquot was injected into the LC–MS/MS system.

### 2.3. Pharmacokinetic Study

Male ICR mice (7-weeks-year-old, 30–35 g) were purchased from Samtako Co. (Osan, Kyunggi-do, Korea). Animals were acclimatized for 1 week in an animal facility at Kyungpook National University. Food and water were available ad libitum. The overall experimental scheme is shown in Figure 2.

#### 2.3.1. The Effect of Amoxicillin on the Intestinal Metabolism and the Pharmacokinetics of Ginsenosides

To investigate the effect of amoxicillin on the intestinal metabolism of ginsenosides, mice received amoxicillin (*n* = 9, 20 mg/kg/day, dissolved in water at 2 mL/kg, twice daily) for 3 days via oral gavage. Before blood sampling, mice were anesthetized using isoflurane (isoflurane vaporizer to 2% with oxygen flow at 0.8 L/min) for 5 min. Blood sampling was performed using a sparse sampling method via the right or left retro-orbital vein under isoflurane anesthesia at 24 and 48 h through the heparinized capillary tube (Heinz Herenz, Hamburg, Germany). The last blood sampling was performed via abdominal artery using a heparin-treated 1-mL syringe (Jung Lim Co. Ltd., Choong-Buk, Korea) under anesthesia with isoflurane at 72 h after the first dose of amoxicillin solution (time schedule and blood sampling volume are given in Table 3). After the centrifugation of the blood samples at 10,000× *g* for 1 min, 30 μL aliquots of plasma were stored at −80 °C for the analysis of plasma amoxicillin concentration. Aliquots (200 μL) of an IS (0.05 ng/mL berberine in acetonitrile) were added to 30 μL of plasma samples. After that, the mixture was vortexed for 15 min and centrifuged at 16,000× *g* for 5 min. After centrifugation, 200 μL of the supernatant was transferred to a clean tube and dried under a nitrogen stream at 40 °C. The residue was reconstituted using 100 μL of 50% acetonitrile supplemented with 0.1% formic acid, and a 2-μL aliquot was injected into the LC–MS/MS system.

Among the amoxicillin treated mice, six mice received RGE in a single dose (2 g/kg suspended in water at 2 mL/kg) via oral gavage 2 h after the last amoxicillin administration on 4th day and then returned to their metabolic cages with food and water ad libitum and urine and feces samples were collected for 48 h. The urine and feces samples were weighed, and 30 μL aliquots of urine and 100 μL aliquots of 10% feces homogenates were stored at −80 °C until the analysis of the ginsenosides. Blood sampling was performed using a sparse sampling method via the right or left retro-orbital vein under anesthesia with isoflurane at 0, 2, 4, and 8 h after the RGE administration through the heparinized capillary tube. The last blood sampling was performed via abdominal artery using a heparin-treated 1-mL syringe under isoflurane anesthesia at 24 and 48 h after the RGE administration (time schedule and blood sampling volume was given in Table 3). After the centrifugation of the blood samples at 10,000× *g* for 1 min, 30-μL aliquots of plasma were stored at −80 °C until the analysis of the ginsenosides. For the comparison, six mice received water (2 mL/kg) via oral gavage for 3 days and, on 4th day, mice received RGE in a single dose (2 g/kg suspended in water at 2 mL/kg) via oral gavage 2 h after the last water administration and then returned to their metabolic cages with food and water ad libitum and urine and feces samples were collected for 48 h.

Aliquots (200 μL) of an IS (0.05 ng/mL berberine in methanol) were added to 30 μL of plasma or urine samples. Aliquots (600 μL) of an IS methanol solution containing 0.05 ng/mL berberine were added to 100 μL of 10% feces homogenate samples. After that, the mixture was vortexed for 15 min and centrifuged at 16,000× *g* for 5 min. After centrifugation, 200 μL of the supernatant was transferred to a clean tube and dried under a nitrogen stream at 40 °C. The residue was reconstituted using 100 μL of 70% methanol supplemented with 0.1% formic acid, and a 10-μL aliquot was injected into the LC–MS/MS system.

#### 2.3.2. The Effect of LAB Supplementation on the Pharmacokinetics of Ginsenosides in Amoxicillin Treated Mice

Mice received amoxicillin (*n* = 12, 20 mg/kg, dissolved in water at 2 mL/kg, twice daily) for 3 days via oral gavage. From the 4th day, mice from the LAB + amoxicillin treatment group received LAB (2 g/kg suspended in water at 2 mL/kg, once daily) for 7 days via oral gavage. Mice from the amoxicillin treatment group received water (2 mL/kg, once daily) for 7 days via oral gavage. Subsequently, on the 10th day, mice received RGE in a single dose (2 g/kg suspended in water at 2 mL/kg) via oral gavage 2 h after the last LAB or water administration and then returned to their metabolic cages with food and water ad libitum and urine and feces samples were collected for 48 h. The urine and feces samples were weighed, and 30 μL aliquots of urine and 100 μL aliquots of 10% feces homogenates were stored at −80 °C until the analysis of the ginsenosides.

Blood sampling was performed using a sparse sampling method via the right or left retro-orbital vein under isoflurane anesthesia at 0, 2, 4, and 8 h after the RGE administration through the heparinized capillary tube. The last blood sampling was performed via abdominal artery using heparin-treated 1 mL syringe under isoflurane anesthesia at 24 and 48 h after the RGE administration (time schedule and blood sampling volume was given in Table 4). After centrifugation of the blood samples at 10,000× *g* for 1 min, 30 μL aliquots of plasma were stored at −80 °C until the analysis of the ginsenosides. Subsequent protocols were identical to the amoxicillin treated group.

### 2.4. Plasma Protein Binding of Ginsenosides

The protein binding of 15 ginsenosides, GRb1, GRb2, GRc, GRd, GRe, GRf, GRg1, GRg3, GRh1, GRh2, GF1, GF2, CK, PPD, and PPT (1 μM each), in pooled mouse plasma (purchased from Sigma-Aldrich, St. Louis, MO, USA) was determined using a rapid equilibrium dialysis kit (ThermoFisher Scientific Korea, Seoul, Korea) according to the manufacturer’s instructions. Briefly, 100 μL of mouse plasma containing 1 μM of each ginsenoside was added to the semipermeable membrane’s inner sample chamber (molecular weight cut-off 8000 Da), and 300 μL of PBS was added to the outer buffer chamber. The samples were then incubated for 4 h at 37 °C on a shaking incubator at 300 rpm, followed by collecting 30-μL aliquots from both the sample and buffer chambers. Samples were mixed with equal volumes of fresh PBS or blank mouse plasma to match the sample matrices and aliquots (200 μL) of an IS (0.05 ng/mL berberine in methanol) were added to 60 μL of matrix matched samples. The mixture was vortexed for 15 min and centrifuged at 16,000× *g* for 5 min. After centrifugation, a 2 μL aliquot was injected into the LC–MS/MS system to analyze ginsenoside concentration.

Positive control study using atenolol (for low protein binding) and propranolol (for high protein binding) was also performed. A 100 μL aliquot of mouse plasma containing 1 μM of atenolol or propranolol was added to sample chamber and 300 μL of PBS was added to the outer buffer chamber. Subsequent protocols were identical to the method described above except for the use of ice-cold IS solution (200 μL; 0.05 ng/mL berberine in acetonitrile) to 30 μL of the reaction samples.

### 2.5. Plasma and Microsomal Stability of Ginsenoside

The plasma stability of 15 ginsenosides, GRb1, GRb2, GRc, GRd, GRe, GRf, GRg1, GRg3, GRh1, GRh2, GF1, GF2, CK, PPD, and PPT (1 μM each), in pooled mouse plasma (purchased from Sigma-Aldrich, St. Louis, MO, USA) and in pooled liver microsomes prepared from male CD-1 mice (Corning Life Sciences; Tewksbury, MA, USA) was determined. For plasma stability, 100 μL of mouse plasma containing 1 μM of each ginsenoside was incubated for 2 h at 37 °C on a shaking incubator at 300 rpm, this followed by collecting 30-μL aliquots from the incubation tubes. The reaction was quenched by addition of an ice-cold IS solution (200 μL; 0.05 ng/mL berberine in methanol) to 30 μL of plasma samples. Thereafter, the mixture was vortexed for 15 min and centrifuged at 16,000× *g* for 5 min. After centrifugation, a 2-μL aliquot was injected into the LC–MS/MS system.

For microsomal stability, individual ginsenoside (1 μM each) was reconstituted in 100 mM potassium phosphate buffer (pH 7.4) containing 0.25 mg of mouse liver microsomes and preincubated for 5 min at 37 °C. This reaction was initiated by adding an NADPH-generating system (1.3 mM β-NADP, 3.3 mM glucose-6-phosphate, 3.3 mM MgCl_2_, and 1.0 unit/mL glucose-6-phosphate dehydrogenase; purchased form Corning Life Sciences) (to make final volume of 100 μL). The reaction mixture was incubated for 1 h at 37 °C in a shaking water bath and the reaction was quenched by addition of addition of an ice-cold IS solution (200 μL; 0.05 ng/mL berberine in methanol) to 30 μL of the reaction samples. The mixture was vortexed for 15 min and centrifuged at 16,000× *g* for 5 min. After centrifugation, a 2 μL aliquot was injected into the LC–MS/MS system.

Positive control study using 1 μM metformin (for high microsomal stability) and 1 μM propranolol (for low microsomal stability) was also performed to ensure the system feasibility. Subsequent protocols were identical to the method described above except for the use of ice-cold IS solution (200 μL; 0.05 ng/mL berberine in acetonitrile) to 30 μL of the reaction samples.

### 2.6. Caco-2 Permeability of Ginsenoside

Caco-2 cells (passage no 41–43; purchased from ATCC, Rockville, MD, USA) were grown in tissue culture flasks containing Dulbecco’s modified eagle medium supplemented with 20% fecal bovine serum, 1% non-essential amino acids, and 1% penicillin-streptomycin. Caco-2 cells were seeded on collagen-coated 12-transwell membranes at a density of 5 × 10^5^ cells/mL and maintained at 37 °C in a humidified atmosphere with 5% CO_2_/95% air for 21 days. The culture medium was replaced every other day. On the day of the experiment, the growth medium was discarded, and the attached cells were washed with prewarmed HBSS (pH 7.4) and preincubated with HBSS for 20 min at 37 °C, and the permeability assay was conducted as previously described [53]. Briefly, to measure the apical to basal permeability of each ginsenoside, 0.5 mL of HBSS containing 50 µM of individual ginsenoside (GRb1, GRb2, GRc, GRd, GRe, GRf, GRg1, GRg3, GRh1, GRh2, GF1, GF2, CK, PPD, or PPT) was added to the apical side (inside of the insert) and 1.5 mL of fresh HBSS was added to the basal side of the insert. The insert was transferred to a well containing 1.5 mL of fresh HBSS every 15 min for 1 h. Aliquots (0.1 mL) in basal side were transferred to clean tubes and stored at −80 °C until further analysis. Transport study was performed for 1 h based on the linearity of the cumulative transport amounts of ginsenosides against the transport time. The integrity of cell monolayers was evaluated before and after the permeability experiments by measuring the transepithelial electrical resistance (TEER) values using an epithelial volt/ohm meter (World Precision Instruments; Sarasota, FL, USA).

To confirm the feasibility of Caco-2 permeability study from apical to basal direction, four permeability marker compounds such as caffeine and propranolol (for high permeability), ofloxacin (for moderate permeability), and atenolol (for low permeability) were used [54]. Briefly, 0.5 mL of HBSS containing the mixture of 2 µM caffeine, 2 µM propranolol, 10 µM ofloxacin, and 50 µM atenolol was added to the apical side and 1.5 mL of fresh HBSS without marker compounds were added to basal side of the insert. Every 15 min, the insert was transferred to a well containing 1.5 mL of fresh HBSS for 1 h. Aliquots (0.2 mL) in basal side were transferred to clean tubes and stored at −80 °C. For the analysis of these compounds, the thawed 200-μL samples were extracted using 200 μL of acetonitrile containing 0.05 ng/mL berberine (IS) and vigorous mixing for 10 min followed by sonication for 5 min and centrifugation at 16,000× *g* for 5 min at 4 °C. After centrifugation, an aliquot (2 μL) was injected into an LC-MS/MS.

### 2.7. LC-MS/MS Analysis

The concentrations of amoxicillin in the mouse plasma samples were analyzed using an Agilent 6470 triple quadrupole LC−MS/MS system (Agilent, Wilmington, DE, USA). Amoxicillin peaks were separated on a Polar RP column (150 × 2.0 mm, 4.0 μm particle size; Phenomenex, Torrance, CA, USA) with mobile phase 50% acetonitrile supplemented with 0.1% formic acid at a flow rate of 0.3 mL/min. Quantification was performed using multiple reaction monitoring (MRM) mode at *m/z* 366.3→ 114.0 for amoxicillin (collision energy (CE) of 20 eV, retention time (T_R_) of 2.5 min) and *m/z* 336.0 320.0 for berberin (IS) (CE 30 eV, T_R_ 4.6 min) in the positive ion mode. The standard calibration curve for amoxicillin was linear in the concentration range of 2–5000 ng/mL for the plasma samples, and the interday and intraday precision and accuracy were <15%.

The concentrations of caffeine, propranolol, ofloxacin, atenolol, and metformin, in the samples were measured simultaneously using an Agilent 6470 Triple Quadrupole LC-MS/MS system with a slight modification of the method of Song et al. [53,54,55,56]. Separation was performed on a Luna CN column (2.0 mm × 150 mm, 5 μm; Phenomenex, Torrance, CA, USA) with mobile phase 50% acetonitrile supplemented with 0.1% formic acid at a flow rate of 0.2 mL/min. Quantification was carried out using MRM mode at *m/z* 195→138 for caffeine (CE 15 eV, T_R_ 2.4 min), *m/z* 260→116 for propranolol (CE 10 eV, T_R_ 3.2 min), *m/z* 267→145 for atenolol (CE 25 eV, T_R_ 2.3 min), *m/z* 362→318 for ofloxacin (CE 15 eV, T_R_ 2.6 min), *m/z* 130→71 for metformin (CE 20 eV, T_R_ 2.2 min), and *m/z* 336.0→320.0 for berberin (IS) (CE 30 eV, T_R_ 4.6 min) in the positive ionization mode.

The concentrations of ginsenosides were analyzed using a modified LC-MS/MS method [30,57] using an Agilent 6470 triple quadrupole LC-MS/MS system (Agilent, Wilmington, DE, USA). The ginsenosides were separated on a Polar RP column (150 × 2.0 mm, 4.0 μm particle size) (Phenomenex, Torrance, CA, USA) with a mobile phase consisting of 0.1% formic acid in water (phase A) and 0.1% formic acid in methanol (phase B) at a flow rate of 0.3 mL/min. The gradient elution used was 70% of phase B for 0–0.2 min, 70–90% (phase B) for 0.2–1.0 min, 90% (phase B) for 1.0–6.5 min, 90–70% (phase B) for 6.5–7.0 min, and 70% (phase B) for 7.0–10.0 min. Quantification was performed using MRM mode in the positive ion mode, according to the previously published methods [30,34,57,58]: *m/z* 1131.6→365.1 for GRb1 (CE 65 eV, T_R_ 3.4 min), *m/z* 1101.6→335.1 for GRb2 and GRc (CE 60 eV, T_R_ 4.2 and 3.2 min), *m/z* 969.9→789.5 for GRd and GRe (CE 50 eV, T_R_ 4.8 and 1.7 min), *m/z* 823.5→365.1 for GRf (CE 55 eV, T_R_ 3.2 min), *m/z* 824.0→643.6 for GRg1 (CE 40 eV, T_R_ 1.8 min), *m/z* 807.5→627.5 for GF2 (CE 40 eV, T_R_ 5.9 min), *m/z* 807.5→365.2 for GRg3 (CE 60 eV, T_R_ 5.8 min), *m/z* 661.5→203.1 for GF1 (CE 40 eV, T_R_ 3.7 min), *m/z* 603.4→423.4 for GRh1 (CE 10 eV, T_R_ 3.2 min), *m/z* 587.4→407.4 for GRh2 (CE 15 eV, T_R_ 6.6 min), *m/z* 645.5→203.1 for CK (CE 35 eV, T_R_ 6.6 min), *m/z* 425.3→109.1 for PPT (CE 30 eV, T_R_ 5.4 min), *m/z* 411.3→109.1 for PPD (CE 25 eV, T_R_ 7.3 min), and *m/z* 336.0→320.0 for berberin (IS) (CE 30 eV, T_R_ 3.5 min). For the 15 ginsenosides, the standard calibration curve for the mixture was linear in the concentration range of 0.5–200 ng/mL, and the inter-day and intra-day precision and accuracy for 15 ginsenosides were <15%.

Berberine was selected as a common IS in this study because berberine showed stable and sensitive peaks after the protein precipitation method using either acetonitrile or methanol for the simultaneous analysis of wide range of ginsenosides, permeability marker compounds, as well as 15 probe substrates and metabolites for drug metabolizing enzymes or transporters [30,57,59].

### 2.8. Data Analysis

Pharmacokinetic parameters were calculated using WinNonlin (version 5.1, Pharsight, Mountain View, CA, USA) by a non-compartmental analysis.

Plasma protein binding was calculated using the following equation [31,60]: $$\mathrm{Plasma}\;\mathrm{protein}\;\mathrm{binding}=\left(1-\frac{\mathrm{Drug}\;\mathrm{concentration}\;\mathrm{in}\;\mathrm{buffer}\;\mathrm{chamber}}{\mathrm{Drug}\;\mathrm{concentration}\;\mathrm{in}\;\mathrm{plasma}\;\mathrm{sample}\;\mathrm{chamber}}\right)\times 100\left(\%\right).$$

Plasma and microsomal stability was determined from the percent of remaining concentration of ginsenosides or marker compounds compared to their initial concentration [53,61].

For the permeability calculation, the transport rate of ginsenosides and marker conpounds was calculated from the slope of the regression line from the mean permeated amounts vs. incubation time plot. The apparent permeability (P_app_) was calculated from the following equation [54,62]:$$\mathrm{Papp}\;\left(10^{-6}\mathrm{cm}/s\right)=\frac{\mathrm{transport}\;\mathrm{rate}\;\left(\mathrm{nmol}/\min\right)}{\mathrm{concentration}\left(\mu M\right)\times \mathrm{area}\left({\mathrm{cm}}^{2}\right)\times 60s}.$$

The data are expressed as the means ± standard deviation for the groups. Statistical analysis was performed using the Student *t*-test.

## 3. Results

### 3.1. LAB-Mediated Ginsenosides Metabolism

To ensure the metabolic activity of LAB about ginsenosides, we conducted a fermentation test of RGE with LAB in vitro. Some ginsenoside metabolites, such as CK, PPD, and PPT, which are not contained in natural RGE and final metabolites of ginsenosides, were significantly increased after incubating with LAB (Figure 3A,C). GRh1 and GRg3 also increased about 30% by fermentation with LAB for 7 days and the apparent amount of GRh2 remained unchanged (Figure 3A). Other intermediate metabolites, GRd, GF2, GRg1, and GRf, showed a gradual decrease by fermentation with LAB for 7 days. GRb1, GRb2, GRc, and GRe that are tetraglycosylated PPD-type or triglycosylated PPT-type ginsenoside, and GF1, which is an intermediate metabolite of PPT-type ginsenoside, showed the most significant decrease during the LAB fermentation process (Figure 3A,B). The results suggested that the metabolic activity of PPD-type and PPT-type ginsenosides by incubation with LAB may be different depending on the number of sugar moiety and glycosylated status. Microbial growth was stably maintained for the incubation period (7 days) (Figure 3D).

### 3.2. Effect of Amoxicillin on the Metabolism of Ginsenoside

To investigate the effect of amoxicillin on the pharmacokinetics and metabolism of ginsenosides, we orally administered RGE (2 g/kg) in mice following repeated administration of amoxicillin. It is reported that the concentration of amoxicillin of at least 0.25 μg/mL to a maximum of 2 μg/mL must be maintained to inhibit the growth of LAB [50]. Thus, we verified the amoxicillin concentration in plasma following repeated administration of amoxicillin for three days to ensure the inhibitory effect of amoxicillin on LAB (Figure 4A). The concentration was maintained in the range of 0.9–1.8 μg/mL at 24, 48, and 72 h after the beginning of amoxicillin treatment, which was thought to be enough to impede the growth of LAB (Figure 4B).

During this time, the recovery of ginsenosides from the urine and feces for 48 h was monitored (Figure 4C). The urinary recovery of all detected ginsenosides was much lower than the fecal recovery, which is attributed to the low oral absorption of ginsenosides and their elimination route. That is, the fecal recovery showed the sum of the absorbed and unabsorbed ginsenosides following oral intake of RGE. Additionally, several ginsenosides are reported to favor biliary excretion than renal excretion [7,31,37]. Compared to several ginsenosides identified in urine (i.e., GRb1, GRb2, GRc, GRd, CK, GRe, and GRg1), 15 ginsenosides were detected in feces (Figure 4C,D). By the amoxicillin treatment, tri- or tetraglycosylated ginsenosides (i.e., GRb1, GRb2, GRc, GRg3, GRe, GRf, and GRg1) were increased, whereas mono- or deglycosylated ginsenosides (i.e., F2, CK, PPD, and PPT) were significantly decreased (Figure 4D). These results suggested that the ginsenoside metabolism mediated the gut microbiota seemed to be significantly inhibited by the amoxicillin treatment since antibiotics could inhibit gut microbiota.

### 3.3. Plasma Concentrations of Ginsenosides Following Single Administration of RGE after Repeated Administration of Amoxicillin

Subsequently, we measured plasma ginsenosides concentrations following repeated administration of amoxicillin, and the pharmacokinetic parameters were compared between control and amoxicillin treatment (Figure 5 and Table 5). Plasma concentrations of Rb1, Rb2, and Rc were not significantly affected by the amoxicillin treatment compared to the control group. However, the plasma concentrations and C_max_ value of Rd were decreased in the amoxicillin group compared to the control group (Figure 5 and Table 5), suggesting the decrease of Rd formation in the intestine mediated by intestinal microbiota because of the amoxicillin treatment. Ginsenosides CK, PPD, and PPT, which are not found in natural RGE, and final metabolites of ginsenosides, were detected in the control group. However, they were not seen in the plasma samples from the amoxicillin treatment group (Figure 5). The results suggested that the formation of CK, PPD, and PPT was blocked by the amoxicillin treatment.

### 3.4. Plasma Concentrations of Ginsenosides Following Single Oral Administration of RGE after Repeated Oral Administration of Amoxicillin with or without Repeated LAB Treatment

To investigate how much the gut microbiota-mediated ginsenoside metabolic activity is repaired after LAB administration, we administered amoxicillin for 3 days followed by with or without LAB supplementation for 1 week and compared the ginsenoside pharmacokinetics between two groups (Figure 6A). The recovery of ginsenosides from urine and feces for 48 h has also been monitored. As shown in Figure 6B, urinary excretion of Rd was increased in the Amoxcillin + LAB group compared with that in the amoxicillin group, which reflects the increased plasma concentrations of Rd. The urinary excretion of other ginsenosides was much lower than the fecal recovery and was not significantly affected by the LAB. In the feces (Figure 6C), the recovery of GRb1, GRb2, GRc, GRd, GRg3, GRe, GRf, and GRg1 was significantly decreased by LAB treatment. Considering these ginsenosides were glycosylated ginsenosides, the repeated administration of LAB decreased these ginsenosides by processing glycosylation. As results, their deglycosylated metabolite ginsenosides, such as CK, PPD, and PPT were significantly increased. The intermediated GRh2, GF2, GRh1, and GF1 remained similar in both groups. Moreover, we found that final metabolites, such as CK, PPD, and PPT, which were decreased in fecal excretion after amoxicillin administration due to inhibited metabolism, were much more or similarly repaired in both fecal excretion and plasma concentration after additional LAB treatment as a result of resumed metabolism compared with the amoxicillin group.

Among the PPD-type ginsenosides, GRb1, GRb2, GRc, GRd, CK, PPD, and GRg3 were detected in mouse plasma samples following repeated LAB administration in addito to amoxicillin treatment. Especially, GRg3 was not detected in control mice and amoxicillin group. The results suggested that LAB treatment may increase the absorption or production of GRg3. However, PPT was only detected among the PPT-type ginsenosides (Figure 7). When compared the pharmacokinetic parameters, all parameters of GRb1, GRb2, and GRc from amoxicillin and LAB treatment were not different compared with amoxicillin group. The AUC and C_max_ value of Rd was significantly increased by the LAB treatment without reaching statistical significance in T_max_, MRT, and half-life of GRd by LAB treatment (Table 6). CK, PPD, and PPT, deglycosylated metabolite ginsenosides that were not detected in amoxicillin treament (Figure 6), were all detected by the LAB supplementation, suggesting the role of LAB in the deglycosylated metabolism of ginsenoside.

### 3.5. LAB-Mediated Metabolism of Ginsenoside

To explain the LAB-mediated metabolism of ginsenoside and the resultant pharmacokinetic features, we incubated individual ginsenoside with LAB and calculated the formation rate of ginsenoside (Figure 8). Incubation of tri- or tetraglycosylated-glycosylated PPD-type ginsenosides, GRb1, GRb2, GRc, and GRd with LAB produced GRg3 with the highest formation rate. Further metabolism from GRg3 to GRh2 and PPD was negligible after the incubation for 1 week. However, GRh2 and PPD were produced from Rg3 when incubating GRg3 with LAB. The results suggested the slow metabolism or losing deglycosylation activity in a week. The formation rate of CK from the incubation of GF2 and the formation of PPD from GRh2 were the highest among the ginsenoside formation rate and they were much higher than the formation rate of GF2 or CK from GRd or the formation rate of GRh2 or PPD from GRg3 (Figure 8A). The results suggested that the deglycosylation mediated by LAB was preferable at the C3 site of the ginsenoside structure [31] and this process might be mainly mediated by Lactobacillus rhamnosus and *L. plantarum* considering the involvement of these species and the composition of LAB formulation (content of *L. rhamnosus* and *L. plantarum* 59%) [32,63,64]. Bifidobacterium longum (content 6%) and Enterococcus sp. (content 6%) also have been reported to be involved in the deglycosylaiton of PPD-type ginsenosides [64,65].

Similarly, the incubation of diglycosylated PPT-type ginsenosides, GRe, GRf, and GRg1 with LAB produced GRh1. Among these metabolisms, the formation rate of GRh1 from GRg1 showed the highest formation rate. Further metabolism to GF2 or PPD was also negligible. However, the greater formation rate of PPT from GF1 or GRh1 and the greater formation rate of GRh1 from GRg1 than the formation rate of GF1 suggested the preferable formation of PPT from GRh1 (i.e., deglycosylation at C20 site of ginsenoside structure [31]), mediated by LAB (Figure 8B). Bacteroides sp. and Bifidobacterium sp. are mainly involved in the deglycosylaiton of PPT-type ginsenosides [66]. Additonally, the results also suggested the sequential and stepwise deglycosylation metabolism of ginsenoside by LAB either in PPD-type or PPT-type.

### 3.6. Permeability, Protein Binding, and Stability of Ginsenosides

In summary, plasma exposure of GRd, GRg3, GF2, CK, PPD, and PPT was increased by the repeated administrations of LAB following amoxicillin pretreatment. However, GRh1 and GRh2 were not found in the mouse plasma despite of high formation rate from Rg1, and GF2/GRg3, as well as considerable amount of GRh1 was detected in the feces. To explain the limited existence of GRh1 and GRh2 in mouse plasma, we measured the apparent A to B permeability (P_app,AB_), plasma protein binding, plasma stability, and microsomal stability of 15 ginsenosides (Figure 9) since these properties affect the pharmacokinetics of ginsenosides in addition to their gut metabolism. For the feasibility of our system, positive control studies using marker compounds for Caco-2 permeability, plasma protein binding, and microsomal stability were performed. P_app,AB_ values of caffeine and propranolol (marker for high permeability) were 29.65 ± 3.77 × 10^−6^ cm/s and 21.60 ± 2.35 × 10^−6^ cm/s, respectively, similar to the reference values (18.8–33.1 × 10^−6^ cm/s). P_app,AB_ values of ofloxacin and atenolol, moderate and low permeable marker, respectively, were 5.10 ± 0.71 × 10^−6^ cm/s and 0.46 ± 0.14 × 10^−6^ cm/s, respectively, similar to the reference values (5.3–8.8 × 10^−6^ cm/s and 0.38–0.45 × 10^−6^ cm/s, respectively) in previous papers [62,67,68,69]. Plasma protein binding of atenolol and propranolol was 5.6 ± 0.2% and 89.7 ± 0.2%, respectively, similar to the reference values (lee than 5% and 87–96%, respectively [70,71]. The percent of remaining metformin and propranolol after 1 h incubation in the mouse liver microsomes was 69.35 ± 3.00% and 17.87 ± 2.33%, respectively, which was similar to the previous results [53,55].

As shown in Figure 9A, PPT seemed to be moderately permeable in Caco-2 cells with P_app,AB_ value of 3.72 ± 0.56 × 10^−6^ cm/s, compared with the P_app,AB_ value of ofloxacin. The permeability of the other 14 ginsenosides was below 0.3 × 10^−6^ cm/s, suggesting the low permeability of 14 ginsenosides except for PPT. Nine PPD-type ginsenosides seem to be more permeable as they are deglycosylated but the P_app,AB_ value of PPT-type were not correlated with the number of sugar (Figure 9A right panel). Even the lowest permeable tetraglycosylated ginsenosides GRb1, GRb2, and GRc were found in the plasma (Figure 5 and Figure 7). Average TEER values before and after the transport study were 394 ± 64 Ω·cm^2^ and 376 ± 44 Ω·cm^2^, respectively and the alterations were less than 10% in all cases.

The plasma protein binding of nine PPD-type ginsenosides was over 99% and unaltered depending on the sugar number. However, the plasma protein binding of 6 PPT-type ginsenosides differed depending on their glycosylation states (Figure 9B). More specifically, the amount of plasma protein binding of triglycosylated PPT-type GRe, GRf, and GRg1 was approximately 65–80%. The protein-bound fraction increased with increasing deglycosylation, as the protein binding of GF1 and PPT increased to over 99% (Figure 9B right panel).

When monitoring the plasma stability of 15 ginsenosides, all ginsenosides were stable from the incubation of these ginsenosides in the mouse plasma for 2 h (Figure 9C). Highly glycosylated ginsenosides with over the sugar number of two were stable from the incubation of these ginsenosides in the mouse liver microsomes for 1 h (Figure 9D). The monoglycosylated or deglycosylated ginsenosides, GRh2, PPD, GRh1, and PPT were unstable from the mouse liver microsomal incubation for 1 h (Figure 9D).

## 4. Discussion

Even though the involvement of LAB in the ginsenoside metabolism has been reported, the effect of LAB on plasma concentrations of ginsenosides has not been investigated in vivo. To minimize the microbiota-mediated metabolism in the gut, we pretreated amoxicillin, which is susceptible to probiotic *Lactobacillis* and *Bifidobacterium* species [50,51,52], for 3 days. The amoxicillin pretreatment maintained the concentrations of amoxicillin in the range of 0.9–1.8 μg/mL for 72 h, which could change the ginsenoside metabolism in the gut. The recovery of GF2, CK, PPD, and PPT from the feces after single oral administration of RGE (2 g/kg) was significantly decreased by the amoxicillin treatment, suggesting the blockade of gut microbiota-mediated deglycosylation of ginsenoside. As a result, the amount of ginsenosides with a sugar number of one or zero (i.e., CK, GRh2, PPD, GF1, and PPT) was decreased, but ginsenosides with a sugar number of more than three were increased (Figure 4D). Plasma concentrations of ginsenosides reflected these alterations. Consistent with the fecal recovery, the plasma exposure of CK, PPD, and PPT was not detectable after the amoxicillin pretreatment (Figure 5). These results suggested that alterations in the gut microbiota environment could change the ginsenoside metabolism and plasma concentrations of ginsenosides. In addition, LAB supplementation showed much higher AUC values of PPD and PPT (1607.3 ± 463.7 ng/mL·h for PPD and 894.4 ± 161.3 ng/mL·h for PPT) compared with control groups (608.7 ± 133.8 ng/mL·h for PPD and 547.9 ± 77.6 ng/mL·h for PPT) (*p* < 0.05). The high and stable plasma concentrations of PPD and PPT in the LAB supplemented group can be explained by the more efficient metabolic conversion from CK or GRh2 to PPD and from GRh1 to PPT (Figure 1) compared with intrinsic activity mediated by gut microbiota. Considering the therapeutic benefits of deglycosylated ginsenosides such as GRg3, CK, PPD, and PPT, the pharmacokinetic alteration of ginsenosides depends on the glycosylation states by the LAB supplementation need further investigation in relation to their pharmacological activities.

The appearance of GRg3 in mouse plasma following LAB supplementation (Figure 7) could be attributed to the preferable formation of GRg3 from GRb1, GRb2, GRc, and GRd that are the most abundant ginsenosides in RGE in addition to the great increase in CK, PPD, and PPT from the incubation of RGE with LAB (Figure 3 and Figure 8). Consistent with this, GRg3 was detected in mouse plasma samples following repeated administrations of RGE (2 g/kg/day) for 14 days [31]. Kim et al. [72] reported that the plasma AUCs of GRd, GRg3, GF2, and CK were correlated with the intake amount of these ginsenosides in humans. Additionally, plasma concentrations of GRg3 after the oral intake of black ginseng, which contains a higher amount of GRg3 than RGE, were significantly higher than that after oral intake of RGE in human [73]. Contrary to the case of GRg3, GRh1 and GRh2 was not detected in the plasma from mice supplemented with LAB. The metabolic instability of GRh2 and GRh1 in mouse liver microsomes (Figure 9D) could explain the limited appearance of these ginsenosides despite of the high formation rate of GRh2 and GRh1 from the LAB incubation (Figure 8) as well as the comparable permeability, plasma protein binding, and plasma stability compared with GRg3 (Figure 9). We also should note that most PPD-type ginsenosides except for GRh2 appeared in mouse plasma, whereas only PPT was detected in mouse plasma following the oral intake of RGE followed by LAB supplementation. It suggested difference in absorption, distribution, excretion mechanism between PPD-type and PPT-type ginsenosides that need further investigation. The high plasma protein binding (Figure 9B), low hepatic distribution (i.e., Ratio of liver AUC to plasma AUC < 0.2 [60]), metabolic and plasma stability (Figure 9C,D) of GRb1, GRb2, GRc, and GRd could coordinately contribute to the long plasma circulation time of these ginsenosides. Jiang et al. [74] reported the substrate specificity for organic anion transporting polypeptide (OATP)1B3 of highly glycosylated PPT-type GRe and GRg1. In addition to the low plasma protein binding (Figure 9B), the substrate specificity to OATP1B3 could account for its fast elimination of GRe and GRg1 (Figure 6B). In our urine samples, GRe and GRg1 were detected after oral administration of RGE (Figure 4C and Figure 6B), suggesting the absorption into plasma and fast elimination process of these ginsenosides. Contrary to these highly glycosylated ginsenosides, metabolic instability of GRg3, GRh1, GRh2, CK, PPD, and PPT (Figure 9D) could facilitate the elimination of GRg3, CK, PPD, and PPT from the plasma. Consistent with this, PPD and PPT were reported to be subjected to hepatic cytochrome P450 (P450)-catalyzed metabolism in liver microsomes [75].

## 5. Conclusions

In this study, we first reported the effect of LAB supplementation on the formation rate of deglycosylated gensenosides, such as CK, PPD, and PPT and on the pharmacokinetics of these deglycosylated ginsenosides in mice. In vitro fermentation of RGE with LAB indicated the highest formation rate of CK, PPD, and PPT that are not in RGE but pharmacologically active metabolites of ginsenosides. This could be explained by the preferable and sequential deglycosylation process of PPD- or PPT-type ginsenosides (i.e., GF2 → CK, GRg3 → GRh2 → PPD, and GRh1 → PPT). In conclusion, LAB supplementation through the repeated administration of LAB for 1 week after the amoxicillin treatment restored the ginsenoside metabolism and recovered the pharmacokinetics of ginsenosides in mice.

## Figures and Tables

**Figure 1 pharmaceutics-13-01496-f001:**
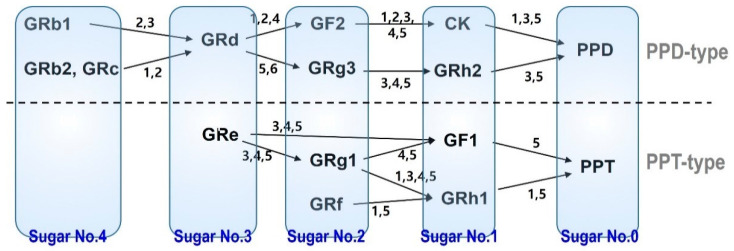
PPD- and PPT-type ginsenosides were grouped by the sugar number and the metabolic pathway of ginsenosides related to gut microbiota. 1: *Bifidobacterium* sp.; 2: *Lactobacillus* sp.; 3: *Eubacterium* sp.; 4: *Fusobacterium* sp.; 5: *Bacteroides* sp.; 6: *Microbacterium* sp.

**Figure 2 pharmaceutics-13-01496-f002:**
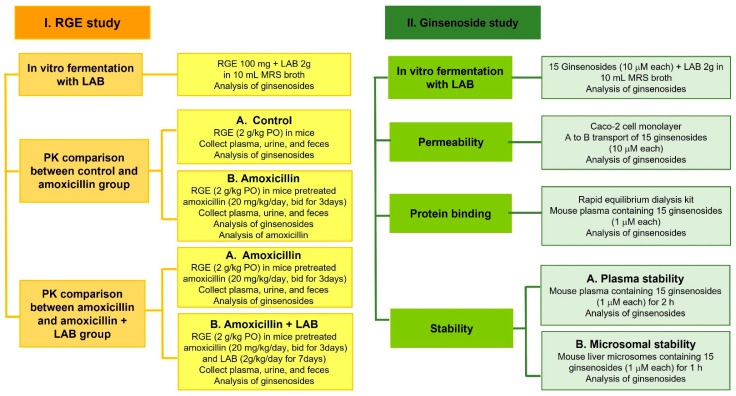
Overall experimental scheme for studies using RGE and 15 ginsenosides.

**Figure 3 pharmaceutics-13-01496-f003:**
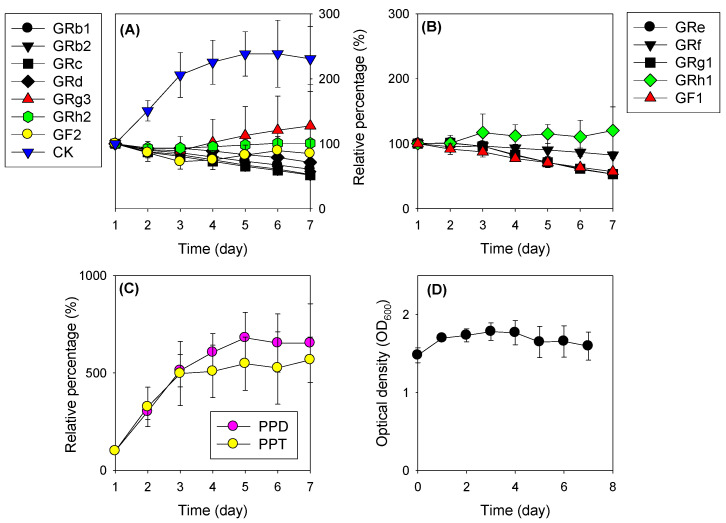
Alterations of 15 ginsenosides following the incubation of RGE with LAB for 7 days were expressed as the relative percentage of ginsenoside on the first day. (**A**) PPD-type ginsenosides except for final metabolite PPD; (**B**) PPT-type ginsenosides except for final metabolite PPT; (**C**) final metabolite ginsenosides PPD and PPT; (**D**) Optical density at 600 nm (OD_600_) of the MRS broth incubating RGE with LAB for 7 days. Data point represents the mean ± standard deviation (*n* = 3).

**Figure 4 pharmaceutics-13-01496-f004:**
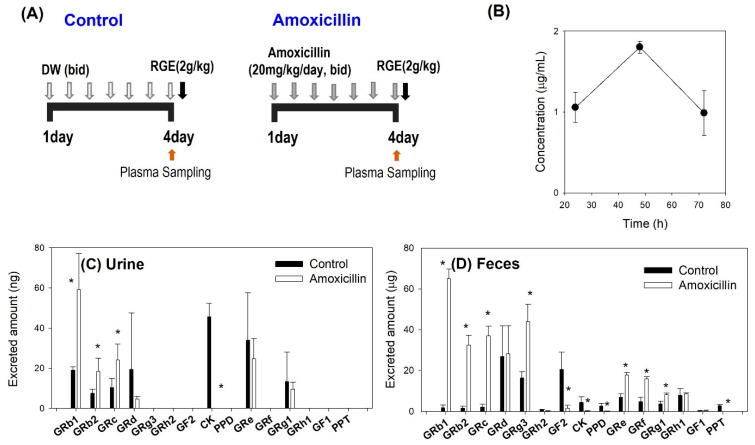
(**A**) Dosing schedule for repeated oral administration of amoxicillin (20 mg/kg/day) and a single oral administration of RGE (2 g/kg). DW: distilled water; bid: twice a day. (**B**) Plasma amoxicillin concentrations after repeated oral administration of amoxicillin for 3 days. Recovery of ginsenosides from (**C**) urine and (**D**) feces for 48 h in mice following a single oral administration of RGE (2 g/kg) after repeated oral administration of amoxicillin for 3 days. The data point represents the mean ± standard deviation (*n* = 3). * *p* < 0.05 compared with control group.

**Figure 5 pharmaceutics-13-01496-f005:**
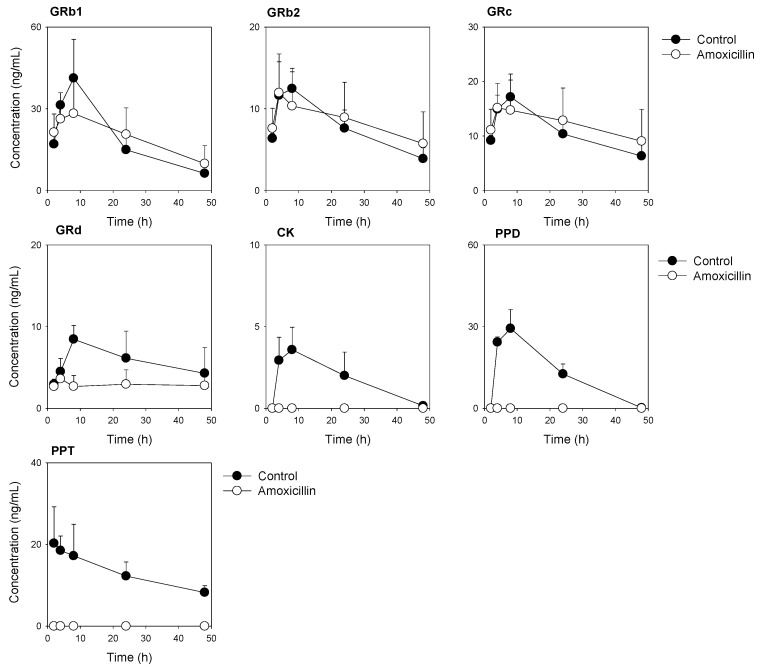
Plasma concentration–time profile of ginsenoside Rb1 (GRb1), GRc, GRd, CK, PPD, and PPT in mice following single oral administration of RGE (2 g/kg) with or without amoxicillin treatment. Data point represents the mean ± standard deviation (*n* = 3).

**Figure 6 pharmaceutics-13-01496-f006:**
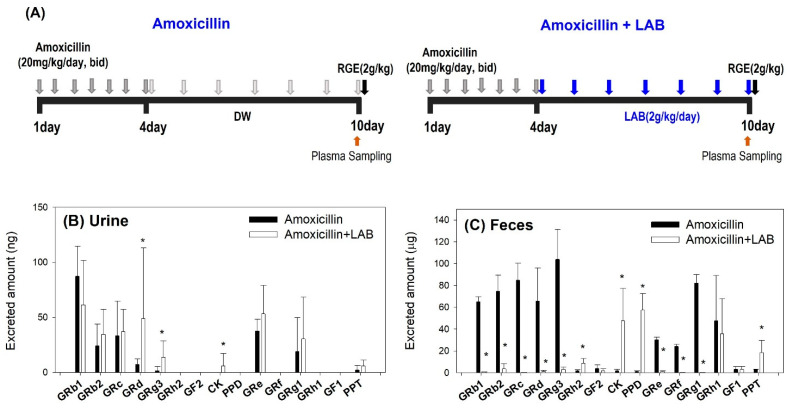
(**A**) Dosing schedule for repeated oral administrations of LAB followed by the amoxicillin (20 mg/kg/day) pretreatment and the subsequent single oral administration of RGE (2 g/kg). DW: distilled water. Recovery of ginsenosides from (**B**) urine and (**C**) feces for 48 h in mice following single oral administration of RGE (2 g/kg) after repeated oral administration of amoxicillin with or without repeated LAB treatment. Data expressed as mean ± standard deviation (*n =* 3). * *p* < 0.05 compared with Amoxicillin group.

**Figure 7 pharmaceutics-13-01496-f007:**
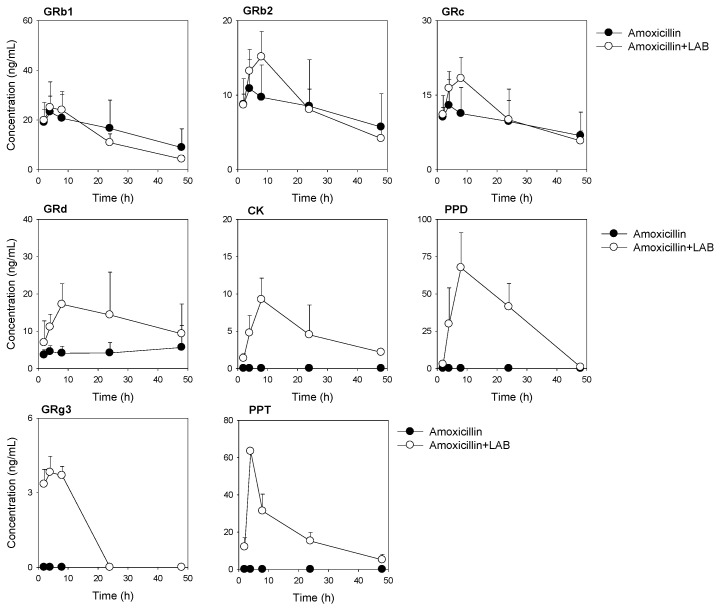
Plasma concentration–time profile of ginsenoside Rb1 (GRb1), GRb2, GRc, GRd, CK, PPD, GRg3, GF2, and PPT in mice following single oral administration of RGE (2 g/kg) after repeated oral administration of amoxicillin with or without repeated LAB treatment. Data point represents the mean ± standard deviation (*n* = 3).

**Figure 8 pharmaceutics-13-01496-f008:**
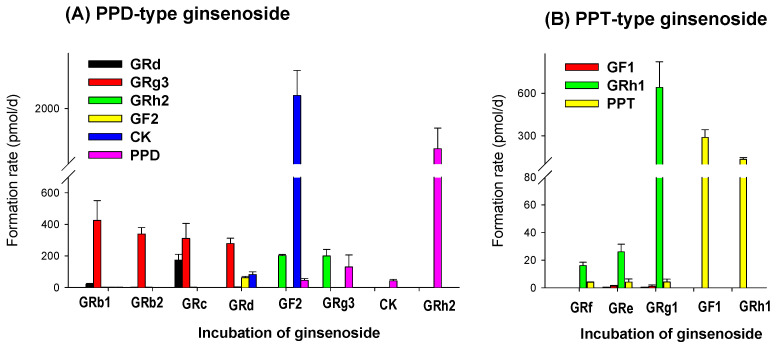
Formation rate of metabolite ginsenoside following the incubation of single ginsenoside with LAB. (**A**) PPD-type ginsenosides: GRb1, GRb2, GRc, GRd, GF2, GRg3, CK, and GRh2 and (**B**) PPT-type ginsenosides: GRf, GRe, GRg1, GF1, and GRh1 were incubated in the presence of LAB for 7 days. Data expressed as mean ± standard deviation (*n* = 3).

**Figure 9 pharmaceutics-13-01496-f009:**
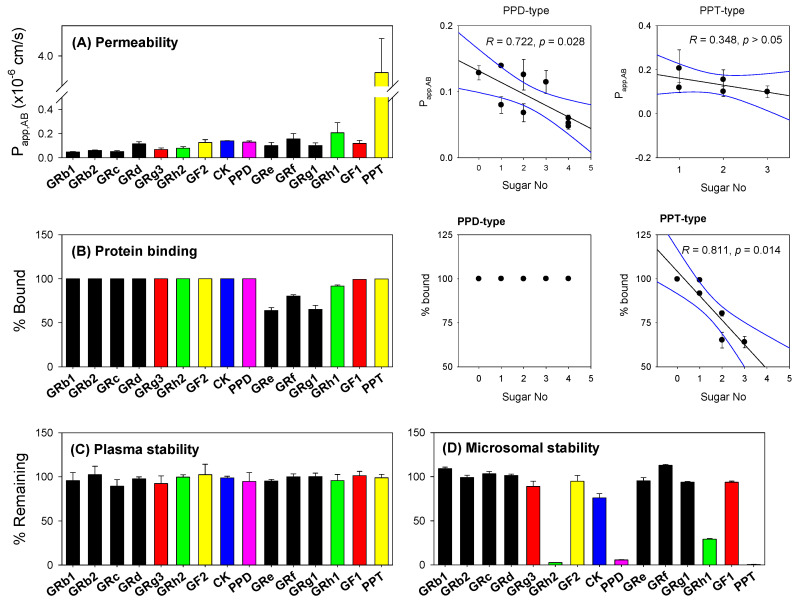
(**A**) The apical to basal permeability (P_app,AB_) of 15 ginsenosides was measured in Caco-2 cell monolayers. Correlation analysis between P_app,AB_ of 9 PPD-type ginsenosides and 5 PPT-type ginsenosides except for PPT (total) and the sugar number of these ginsenosides was shown in the right panel. (**B**) Plasma protein binding of 15 ginsenosides were measured in mouse plasma using a rapid equilibrium kit. Correlation analysis between plasma protein binding (% bound) of 9 PPD-type and 6 PPT-type ginsenosides and the sugar number of these ginsenosides was shown in the right panel. (**C**) Plasma stability of 15 ginsenosides was measured in mouse plasma. (**D**) The metabolic stability of 15 ginsenosides was measured using mouse liver microsomes. Data expressed as mean ± standard deviation (*n* = 3).

**Table 1 pharmaceutics-13-01496-t001:** The content of ginsenoside in red ginseng extract (RGE).

PPD-Type	Content (mg/g RGE)	PPT-Type	Content (mg/g RGE)
GRb1	4.7	GRg3	3.5
GRb2	2.3	GRh2	ND
GRc	2.5	CK	ND
GRd	1.3	PPD	ND
F2	ND	Total content of PPD-type 14.3 mg/g RGE	
PPT-type	Content (mg/g RGE)	PPT-type	Content (mg/g RGE)
GRe	1.3	GRh1	1.6
GRg1	0.6	F1	ND
GRf	1.1	PPT	ND
		Total content of PPD-type 4.6 mg/g RGE	

**Table 2 pharmaceutics-13-01496-t002:** Strains contained in the commercially available lactic acid bacterial formulation (LAB).

Species	Species	Species
*L. helveticus*	*L. rhamnosus*	*B. lactis*
*L. bulgaricus*	*L. casei*	*Enterococcus faecium*
*L. fermentum*	*L. reuteri*	*Enterococcus faecalis*
*L. gasseri*	*L. plantarum*	*Lactococcus lactis*
*L. paracasei*	*L. salivarius*	*Streptococcus thermophiles*
*L. acidophilus*	*B. longum*	
*B. breve*	*B. bifidum*	Total 1 billion CFU/2 g

**Table 3 pharmaceutics-13-01496-t003:** Blood sampling method for the effect of amoxicillin on ginsenoside pharmacokinetics in mice.

Group	Amoxicillin				Control		
Sampling Time (h)	Group 1 (*n* = 3)	Group 2 (*n* = 3)	Group 3 (*n* = 3)	Blood Volume (μL)	Group 4 (*n* = 3)	Group 5 (*n* = 3)	Blood Volume (μL)
0		RO–right		80	RO–right		80
2			RO–right	80		RO–right	80
4		RO–left		80	RO–left		80
8			RO–left	80		RO–left	80
24	RO–right	AA		100	AA		100
48	RO–left		AA	100		AA	100
72	AA			100			

RO–right: retro-orbital blood sampling—right eye under anesthesia with isoflurane. RO–left: retro-orbital blood sampling—left eye under anesthesia with isoflurane. AA: abdominal artery blood sampling under anesthesia with isoflurane.

**Table 4 pharmaceutics-13-01496-t004:** Blood sampling method for the effect of LAB on ginsenoside pharmacokinetic study in mice.

Group	Amoxicillin + LAB			Amoxicillin		
Sampling Time (h)	Group 1 (*n* = 3)	Group 2 (*n* = 3)	Blood Volume (μL)	Group 3 (*n* = 3)	Group 4 (*n* = 3)	Blood Volume (μL)
0	RO–right		80	RO–right		80
2		RO–right	80		RO–right	80
4	RO–left		80	RO–left		80
8		RO–left	80		RO–left	80
24	AA		100	AA		100
48		AA	100		AA	100

RO–right: retro-orbital blood sampling—right eye under anesthesia with isoflurane. RO–left: retro-orbital blood sampling—left eye under anesthesia with isoflurane. AA: abdominal artery blood sampling under anesthesia with isoflurane.

**Table 5 pharmaceutics-13-01496-t005:** Pharmacokinetic parameters of ginsenosides in mouse after single dosing RGE with or without amoxicillin treatment.

Ginsenosides	Control				
	AUC (ng/mL·h)	C_max_ (ng/mL)	T_max_ (h)	MRT (h)	Half-Life (h)
GRb1	895.9 ± 81.3	41.3 ± 14.3	8.0 ± 0.0	16.3 ± 2.7	25.9 ± 4.7
GRb2	361.4 ± 74.3	12.5 ± 2.1	8.0 ± 0.0	19.4 ± 1.6	24.6 ± 6.7
GRc	508.6 ± 109	17.2 ± 3.1	8.0 ± 0.0	19.9 ± 1.7	22.8 ± 6.7
GRd	278.8 ± 117	8.6 ± 1.8 *	13.3 ± 9.2	21.3 ± 3.0	24.2 ± 3.1
CK	69.94 ± 37.8	3.6 ± 1.4	8.0 ± 0.0	13.9 ± 5.3	13.2 ± 3.2
PPD	608.7 ± 133.8	29.3 ± 7.0	8.0 ± 0.0	12.6 ± 4.7	7.9 ± 1.8
PPT	547.9 ± 77.6	20.3 ± 8.9	2.0 ± 0.0	20.2 ± 2.2	20.4 ± 2.3
**Ginsenosides**	**Amoxicillin**				
	**AUC (ng/mL·h)**	**C_max_ (ng/mL)**	**T_max_ (h)**	**MRT (h)**	**Half-life (h)**
GRb1	926.0 ± 439	28.2 ± 12.9	8.0 ± 0.0	19.3 ± 1.4	21.7 ± 3.6
GRb2	391.5 ± 192	10.3 ± 4.6	13.3 ± 9.2	21.4 ± 1.6	25.1 ± 2.7
GRc	571.7 ± 276	14.8 ± 6.6	13.3 ± 9.2	21.9 ± 1.5	25.1 ± 9.9
GRd	132.4 ± 77	3.2 ± 1.6	24.7 ± 23.0	24.0 ± 2.1	29.1 ± 8.6
CK	NC	NC	NC	NC	NC
PPD	NC	NC	NC	NC	NC
PPT	NC	NC	NC	NC	NC

AUC: area under the plasma concentration–time curve from 0 to 48 h; C_max_: maximum plasma concentration; T_max_: time to reach C_max_; MRT: mean residence time. NC: Not calculated. Data expressed as mean ± standard deviation (*n =* 3). * *p* < 0.05 compared with Amoxicillin group.

**Table 6 pharmaceutics-13-01496-t006:** Pharmacokinetic parameters of ginsenosides in mouse after single dosing RGE after repeated oral administration of amoxicillin with or without repeated LAB treatment.

Ginsenosides	Amoxicillin				
	AUC (ng/mL·h)	C_max_ (ng/mL)	T_max_ (h)	MRT (h)	Half-Life (h)
GRb1	752.3 ± 477.6	23.7 ± 11.6	3.3 ± 1.2	18.6 ± 1.6	20.1 ± 2.2
GRb2	385.9 ± 243.6	10.9 ± 5.2	4.0 ± 0.0	20.2 ± 1.7	19.2 ± 2.3
GRc	448.2 ± 268.1	13.2 ± 6.6	4.7 ± 3.1	20.6 ± 1.2	20.2 ± 1.8
GRd	213.1 ± 151.3	6.5 ± 5.1	5.3 ± 2.3	24.1 ± 3.9	29.2 ± 2.7
CK	NC	NC	NC	NC	NC
PPD	NC	NC	NC	NC	NC
GRg3	NC	NC	NC	NC	NC
PPT	NC	NC	NC	NC	NC
**Ginsenosides**	**Amoxicillin + LAB**				
	**AUC (ng/mL·h)**	**C_max_ (ng/mL)**	**T_max_ (h)**	**MRT (h)**	**Half-life (h)**
GRb1	622.1 ± 163.6	26.2 ± 4.6	4.7 ± 3.1	16.9 ± 1.0	16.2 ± 0.9
GRb2	419.2 ± 108.6	15.1 ± 3.4	8.0 ± 0.0	18.1 ± 0.9	21.6 ± 1.7
GRc	524.4 ± 146.6	18.3 ± 4.3	8.0 ± 0.0	19.0 ± 1.3	24.1 ± 2.4
GRd	618.1 ± 385.6 *	18.4 ± 6.9 *	16.0 ± 9.2	20.6 ± 3.7	25.5 ± 3.8
CK	249.8 ± 103.0	9.3 ± 2.9	8.0 ± 0.0	13.7 ± 6.2	12.0 ± 5.0
PPD	1607.3 ± 463.7	73.0 ± 14.6	6.7 ± 2.3	16.0 ± 1.2	18.4 ± 2.2
GRg3	25.3 ± 10.6	4.8 ± 1.9	2.5 ± 1.0	3.5 ± 0.9	5.3 ± 1.1
PPT	894.4 ± 161.3	63.4 ± 0.6	4.0 ± 0.0	13.7 ± 2.9	14.4 ± 3.0

AUC: area under the plasma concentration–time curve from 0 to 48 h; C_max_: maximum plasma concentration; T_max_: time to reach C_max_; MRT: mean residence time. NC: Not calculated. Data expressed as mean± standard deviation (*n* = 3). * *p* < 0.05 compared with control group.

## Data Availability

The data presented in this study are available upon request.

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
