# Peer review of "Effect of Lactic Acid Bacteria on the Pharmacokinetics and Metabolism of Ginsenosides in Mice"

_pharmaceutics, 2021, doi:10.3390/pharmaceutics13091496_

Round 1

Reviewer 1 Report

Thank the authors to provide the evidence of the therapeutic efficacy of red ginseng extract (RGE) and many references describing investigations of ginsenosides related to the efficacy. The authors have correctly answered my queries. I have carefully and critically checked the manuscript and references, and then I could understand the novelty of the current study including the possibility of the interactions between the RGE and LAB in relation to the ginsenoside metabolism.

Author Response

Thank you for the reviewer’s comments and we deeply appreciate the reviewer’s understanding.

Reviewer 2 Report

The manuscript is in line with the currently popular trend in medicine to look how the bacterial supplments affect multiple entities. Tha paper is good with data basicly supporting the conclusions from existing literature, but some important issue exists that I want to point

Introduction:

Definitely too long. Yellow marked parts are of the sope and shoul be re-written to a max of 1 paragraph. In my opinion, =information on the onvolvement of gut microbiota in xenobiotics (incl.drugs) metabolism (pharmacomicrobiomics) shoul replace the above pointed text

Methods

  1. The experiment has several arms thus I would highly recommend adding a scheme on what, when, and where was done?
  2. Did you tested LAB abundance after incubation in invitro *why not?)
  3. How was the recovery from faeces and urine done?

Discussion

  1. I would move the very past paragraph up
  2. Discuss on strains specificity

Author Response

This manuscript is a resubmission of an earlier submission. The following is a list of the peer review reports and author responses from that submission.

Round 1

Reviewer 1 Report

The manuscript is well written and describes a thorough investigation of the significance of the intestinal transformation of ginsenosides for their subsequent absorption. 

Minor comments:

In Abstract line 28 and later line 484: "were also disappeared" is used while the authors probably meant that the compounds in question "were not detectable".

Lines 60 and 61: "...are probably more effective and permeable in the intestinal lumen." is confusing. The effectiveness in the lumen is probably not relevant for the main effects of these compounds while there is no permeability "in the intestinal lumen" but rather "through the intestinal mucosa".

Line 68 and elsewhere: A singular form "ginsenoside" is used where plural might be more appropriate.

Line 80: "LAB are..." instead of "LAB is"

Major comment - Lines 139 – 152 and elsewhere: The maximum total blood sampled from mice of the reported weight should not exceed 300uL in 28 days. Considering the volume of the stored plasma (180 uL) and the likely losses during the sampling, this was exceeded. Furthermore, the chosen sampling method is quite crude, contributes to significant additional blood loss, is not appropriate for so many consecutive samples, and its use should be limited to sampling under general anesthesia or in »non-recovery« experiments. For the described study a canulization should be considered, but it is unfortunately too late for the animals used. Beyond the poor animal welfare status of the presented work, this could also affect the validity of the results. I sincerely hope that such protocols are becoming a thing of the past. The authors clearly state the ethical approval of the experiments was obtained at the institutional level, which simply acknowledges there is significant room for improvement of the animal welfare standards of the institution.

The exact volumes of the blood drawn for individual samplings should be given. The anticoagulant used for obtaining blood plasma is not listed in the text. It must be added or in case it was not used the authors should clearly state that blood serum was used for further analysis.

Major comment - Line 188: The origin of “mouse plasma”  and of “mouse liver microsomes” mentioned is not given in the “materials” or other sections.

Minor comment - Line 216: Chromatographic column manufacturers, country of origin, etc. should be given.

Major comment - Lines 202…, 429… and Figure 8A: There is no mention of TEER or other control of cell monolayer integrity and viability. The concomitant use of multiple tested compounds is known to significantly alter the measured apparent permeability coefficients through yet unknown mechanisms. The apparent permeability values of internal or at least external permeability standards should be given to evaluate the obtained values through comparison. Without such control, it would be better to omit the Caco-2 part of the study altogether. The authors state “PPT seemed to be porous” with which they probably meant that it was highly permeable, but this statement has no validity without a comparison to a known highly permeable model drug measured under identical conditions. It is also worth noting that the sampling, which ended at 60 min would be more appropriate for compounds with much higher permeability while the compounds tested for the manuscript could be well within the lag-time phase of the permeation when the experiments were already finished.

Minor comment - Line 205: "Penicillin-" instead of "Pecicillin".

Line 433: "was existed" should probably be "was detected" or something similar.

Line 535: Perhaps the authors were trying to say "conversion of ginsenosides to PPD and PPT"?

Reviewer 2 Report

This manuscript described the effects of lactic acid bacteria on the pharmacokinetics of ginsenosides in mice treated with red ginseng extract. In my opinion, the priority for publication in Pharmaceutics is low. There are majour comments as described below:

  1. Why did the authors select mice to perform pharmacokinetic study? Blood sampling volume is limited in mice. Blood sampling volume from mice should be described. Generally, in pharmacokinetic study, rat is selected and used. Sparse sampling data cannot correctly determine the inter- and intra-individual variability in PK parameters.

  1. The authors concluded that the supplementation of lactic acid bacteria with oral administrations of red ginseng extract would help increase plasma concentrations of deglycosylated ginsenosides, but there is no information about a relationship between plasma concentration levels and therapeutic efficacies of red ginseng extract. Unfortunately, I cannot find novel issue regarding to pharmacokinetics in mice with or without intestinal bacteria.

Author Response

  1. Why did the authors select mice to perform pharmacokinetic study? Blood sampling volume is limited in mice. Blood sampling volume from mice should be described. Generally, in pharmacokinetic study, rat is selected and used. Sparse sampling data cannot correctly determine the inter- and intra-individual variability in PK parameters.

(Answer) Thank you for the critical but valuable comments. We also agreed the reviewer’s commetns. However, we selected mouse as an experimental animal in this study because the plasma profile of ginsenosides in mice rather than in rats were more similar to those in human based on our previous results. We added this description in the Introduction of the revised manuscript during the revision.

(Page 2-3, lines 81-90) : Jeon et al. [25] reported the species difference in the gut metabolism and absorption of ginsenosides in the pathway from GRd to PPD and from GRe to PPT between mice and rats. The pharmacokinetic features of ginsenosides after repeated oral administration of RGE in humans were similar to mice rather than those of rats. That is, GRb1, GRb2, GRc, GRd, GRg3, CK, PPD, and GRe were detected in mouse plasma but GRb1, GRb2, GRc, GRd, PPD, and PPT were detected in rat plasma after the repeated oral administration of RGE [25]. In human plasma, GRb1, GRb2, GRc, GRd, GRg3, CK, PPD, and PPT were de-tected [9]. Moreover, the half-lives of GRg3, CK, and PPD in mice were similar to those in human subjects [9,25]. Based on these previous results, we selected mice as experimental animals in this study.

  1. The authors concluded that the supplementation of lactic acid bacteria with oral administrations of red ginseng extract would help increase plasma concentrations of deglycosylated ginsenosides, but there is no information about a relationship between plasma concentration levels and therapeutic efficacies of red ginseng extract. Unfortunately, I cannot find novel issue regarding to pharmacokinetics in mice with or without intestinal bacteria.

(Answer) As the reviewer commented, at present there was no relationship between plasma ginsenoside levels and thepapeutic efficacies of red ginseng extract. Therefore we revised the conclution of our manuscript as follows :

(Page 1, lines 29-32) In conclusion, the alterations in the gut microbiota environment could change the ginsenoside metabolism and plasma concentrations of ginsenosides. Therefore, the supplementation of LAB with oral administrations of RGE would help increase plasma concentrations of deglycosylated ginsenosides such as CK, PPD, and PPT.

In addtion, although the involvement of LAB in the deglycosylation of ginsenosides has been adressed in the previous research papers, the in vivo effect of LAB supplements on the pharmacokinetic of ginsenosides was not investigated yet. Therefore, we first reported the effect of LAB supplementation on the formation of deglycosylated gensenosides, such as CK, PPD, and PPT and the pharmacokinetics of these deglycosylated ginsenosides in mice. In addition to the effect of LAB supplementation on the pharmacokinetics of ginsenosides, we also investigated the differential absorptive permeability, protein binding, and metabolic stability of 15 ginsenosides depend on the PPD- or PPT-type and the glycosylation states. We believe that our results would help to understand the food drug interaction between LAB and ginsenosides and the differential pharmacokinetics between PPD-type and PPT-type ginsenosides in mice.

Reviewer 3 Report

Ji-Hyeon Jeon et al investigated the effect of lactic acid bacteria on the metabolism and pharmacokinetics of ginsenosides in ICR mice.

  1. Lines 14-19, the authors need to rephrase the sentences in the abstract.
  2. The manuscript required thorough revision for spelling mistakes and grammar. In table 4 and 5 captions, “sigle” should be modified to “single”. Line 205; not pecicillin, “penicillin”.
  3. The authors should include the anesthesia procedure used for the pharmacokinetic studies.
  4. What would be the reason for conducting the Caco-2 permeability experiment of ginsenosides for only 1h? The permeability direction was not mentioned in the procedure and it needs to be mentioned.
  5. What are the criteria for the selection of internal standards for the LCMS analysis?
  6. The authors should mention the complete pharmacokinetic parameters of ginsenosides including half-life, clearance, the volume of distribution in tables 4 and 5.
  7. The authors need to include the conclusion section in the manuscript.
  8. Figure 1A and Table 3 should be removed from the manuscript and these were already published in the papers “Detection of 13 Ginsenosides (Rb1, Rb2, Rc, Rd, Re, Rf, Rg1, Rg3, Rh2, F1, Compound K, 20(S)-Protopanaxadiol, and 20(S)-Protopanaxatriol) in Human Plasma and Application of the Analytical Method to Human Pharmacokinetic Studies Following Two Week-Repeated Administration of Red Ginseng Extract”; and “Correlation between the Content and Pharmacokinetics of Ginsenosides from Four Different Preparation of Panax Ginseng C.A. Meyer in Rats”. The authors should cite these references and remove Figure 1A and Table 3.
  9. What is the source of mouse liver microsomes? The authors should include the source in the manuscript.

Round 2

Reviewer 1 Report

As I can see the authors have invested great effort in all necessary aspects of the revision in the short time available and have changed the manuscript wherever appropriate and possible. Since it is not possible to retroactively improve the animal welfare, it only makes sense to accept the manuscript in it's current state because repeating the experiments with blood sampling better aligned with current guidelines would unnecessarily require the use of more laboratory animals while the likelihood of obtaining different results is low. I have no further comments which would require major changes to the manuscript.

Reviewer 2 Report

The authors have been adequately answered to my comments. However, I am not yet sure the reason why pharmacokinetic alteration of ginsenosides with or without lactic acid bacteria were investigated, because there was no relationship between plasma ginsenoside levels and therapeutic efficacies of red ginseng extract. The authors should show information that these pharmacokinetic alterations affect its therapeutic efficacies. If not, the backgrounds of the current study is ambiguous. 

Reviewer 3 Report

The authors significantly improved the manuscript and answered all the comments.